# Comparing severe COVID-19 outcomes of first and second/third waves: a prospective single-centre cohort study of health-related quality of life and pulmonary outcomes 6 months after infection

Pernilla Darlington  ,[1,2] Mari Roël,[1,2] Maria Cronhjort,[3] Gabriel Hanna,[4] Anders Hedman,[2,5] Eva Joelsson-Alm,[2,6] Anna Schandl[2,6]

PD and MR are joint first authors.

For numbered affiliations see end of article.

**Correspondence to**
Mrs Pernilla Darlington;
pernilla.darlington@ki.se

## ABSTRACT

**Objective** We aimed to compare long-term outcomes in intensive care unit (ICU) survivors between the first and second/third waves of the COVID-19 pandemic. More specifically, to assess health-related quality of life (HRQL) and respiratory health 6 months post-ICU and to study potential associations between patient characteristic and treatment variables regarding 6-month outcomes.

**Design** Prospective cohort study.

**Setting** Single-centre study of adult COVID-19 patients with respiratory distress admitted to two Swedish ICUs during the first wave (1 March 2020–1 September 2020) and second/third waves (2 September 2020– 1 August 2021) with follow-up approximately 6 months after ICU discharge.

**Participants** Critically ill COVID-19 patients who survived for at least 90 days.

**Main outcome measures** HRQL, extent of residual changes on chest CT scan and pulmonary function were compared between the waves. General linear regression and multivariable logistic regression were used to present mean score differences (MSD) and ORs with 95% CIs.

**Results** Of the 456 (67%) critically ill COVID-19 patients who survived at least 90 days, 278 (61%) were included in the study. Six months after ICU discharge, HRQL was similar between survivors in the pandemic waves, except that the second/third wave survivors had better role physical (MSD 20.2, 95% CI 7.3 to 33.1, p<0.01) and general health (MSD 7.2, 95% CI 0.7 to 13.6, p=0.03) and less bodily pain (MSD 12.2, 95% CI 3.6 to 20.8, p<0.01), while first wave survivors had better diffusing capacity of the lungs for carbon monoxide (OR 1.9, 95% CI 1.1 to 3.5, p=0.03).

**Conclusions** This study indicates that even though intensive care treatment strategies have changed with time, there are few differences in long-term HRQL and respiratory health seems to remain at 6 months for patients surviving critical COVID-19 in the first and second/third waves of the pandemic.

## STRENGTHS AND LIMITATIONS OF THIS STUDY

⇒ The main methodological strengths of the study lie in the prospective and comprehensive data collection of long-term outcomes in critically ill patients with COVID-19.

⇒ The study included well-validated outcome tests and questionnaires.

⇒ Sixty-one per cent of the intensive care unit survivors were included in the study, which may induce a risk of selection bias.

⇒ Further survival rate and follow-up rate were higher for those from the first wave.

⇒ Data on virus strains were not available, so analyses were not adjusted for this parameter.

## INTRODUCTION

COVID-19, caused by SARS-CoV-2, was declared a global pandemic in March 2020 and has resulted in numerous severely ill individuals in need of respiratory support[1] and millions of deaths worldwide.[2] The majority survived the acute episode of a critical illness but were confronted with physical and psychological sequelae after discharge from hospital which prolonged recovery.[3] Approximately 35%–80% of the survivors experienced persistent symptoms of fatigue, dyspnoea, joint pain, sleep disorders, and loss of memory and concentration lasting beyond 4 weeks after a moderate to severe acute infection, which is referred to as 'postacute COVID-19 syndrome'[4–6] Long-lasting respiratory complications as well as other functional deficits which may cause substantial morbidity have been reported among intensive care unit (ICU) survivors in the first year after hospital discharge.[7–10] Furthermore, a high number of

residual abnormalities on chest CT scan have been found several months after discharge from hospital, and acute respiratory distress syndrome has been associated with more severe radiological findings.[11] Since the start of the pandemic, the risk for hospital admission has varied as new variants of SARS-CoV-2 have emerged.[12] Likewise, worldwide different virus variants have been dominating the waves in Sweden.[13] Further, during the second and third waves, scientific and clinical understanding of the illness improved, and potentially more effective treatments were provided. Yet, whether functional outcomes have improved in line with the increase in knowledge about COVID-19 remains unexplored.

We aimed to compare long-term outcomes in ICU survivors between the first and second/third waves of the COVID-19 pandemic. More specifically, to assess health-related quality of life (HRQL) and respiratory health 6 months post-ICU. Moreover, to study potential associations between patient characteristic and treatment variables regarding 6-month outcomes.

## METHODS

### Study design and participants

This was a prospective, single-centre cohort study of patients with acute COVID-19 infection admitted to two ICUs at Södersjukhuset, Stockholm, Sweden between 1 March 2020 and 1 August 2021. All patients who had been critically ill with a positive PCR test for COVID-19 and treated for respiratory failure with invasive ventilation, high-flow treatment with oxygen (HFNO) or non-invasive ventilation (NIV) in the ICU were eligible for inclusion in the study.

### Setting

Before the pandemic, the 2 ICUs had a total number of 16 beds. During the first wave, the number of ICU beds expanded to a maximum of 60 beds in April and May 2020. At the beginning of the second wave, the number of ICU beds were 21, rising to 29 beds in November 2020 and further up to maximum of 33 beds in April and May 2021. At the end of the third wave, the number of beds was 21. The nurse:patient ratio was 1:1–2 and the physician:patient ratio 1:4–6 during all three waves, but the proportion of intensive care specialised nurses and physicians decreased during the peaks of the pandemic. None of the ICUs were exclusive for COVID-19 patients. The admission criteria to the ICU due to COVID-19 were need of respiratory support (ie, HFNO/NIV/invasive ventilation). During the second/third wave, there was a step down for stable patients on HFNO, but still under continues observation. Corticosteroids became standard treatment at the end of the first wave.

### Data collection

A detailed description of this data collection can be found in other publications.[7 8] Data on demographic and clinical details were retrieved from medical charts

regarding age (≤50, 50–65, >65 years), sex (male/female), body mass index (BMI), smoking habits (ever/never), comorbidities (ie, diabetes, hypertension/cardiovascular disease, chronic lung disease) Simplified Acute Physiological Score III (a scoring system used to predict mortality risk in the ICU), where a higher score indicates a higher mortality risk,[14] corticosteroid treatment (yes/no), respiratory support (invasive vs HFNO/NIV) and ICU length of stay.

### Pandemic waves

The study was categorised into two time periods: the first wave (1 March 2020–1 September 2020) and the second/third waves (2 September 2020–1 August 2021). In Sweden, the first cases of severely ill patients were admitted to the ICU at the beginning of March 2020 with a peak in April and then subsided during the summer.[15] Therefore, this time period has been considered to separate the end of the first wave.[16] The beginning of the second wave was estimated to start on 2 September 2020. Thereafter under a longer period, there was an increase in cases with one peak around new year and one peak in April 2021. There was no clear gap in time between the second and third waves, even if there was a clear decrease in cases in March 2021,[15] and the second and third waves were, therefore, grouped as one time period in this study.

### Outcomes

The main outcomes were HRQL and respiratory health assessed approximately 6 months after ICU discharge.

HRQL was assessed with the self-administered RAND-36 Item Health Survey (RAND Corp), a version of The Medical Outcome Study Short Form-36 (SF-36).[17] SF-36 and RAND-36 are commonly used generic instrument for measuring HRQL and rather similar, except for some minor differences, for example, RAND-36 lacks an authorised algorithm for calculation of Mental and Physical Component Summary scores.[18] The Swedish translation (RAND-36 V.1.02) has been shown to be equivalent to the English.[18] It includes reference data for norm-based comparisons and has been shown to be an acceptable and reliable instrument of measuring HRQL in a Swedish general population.[19] The RAND-36 questionnaire includes the following eight domains, namely: physical functioning (10 items), role physical (4 items), bodily pain (2 items), general health (5 items), vitality (4 items), social functioning (2 items), role emotional (3 items) and emotional well-being (5 items).[20] Questionnaire responses were linearly transformed into scores between 0 and 100, where a higher score represents a higher HRQL.[18 20]

Residual changes were identified with a CT scan of the thoracic region using a Siemens Somatom Drive with a standard 120-kV CT thorax protocol as described in a previous publication.[7] International standard thoracic radiological terminology from the Fleischner Society[21] was used to identify the parenchymal changes related to SARS-CoV-2 infection,[22] such as ground glass opacities,

subpleural bands, reticular pattern and bronchiectasis. A semiquantitative CT score (CT involvement score) was estimated based on lobar involvement and then calculated as a total score for all five lobes: 0: 0%, 1: <5%, 2: 5%–25%, 3: 26%–50%, 4: 51%–75%, 5: >75%, yielding a score ranging from 0 to 5 in each lobe and a global score of 0–25.[23]

Pulmonary function was assessed as capacity for gas exchange, that is, diffusing capacity of the lungs for carbon monoxide (DLCO) (uncorrected value), according to standard methods.[24] Results were expressed in accordance with reference values (Hedenström)[25][26] with DLCO<80% as cut-off for impaired pulmonary function.[8][9][11][27]

## Statistical analyses

For comparison of HRQL, multivariable general linear regression models were used to estimate mean score differences (MSD) with 95% CIs. A 3–5 point differences between groups were regarded as minimally clinically important differences.[28] Missing items were accounted for with automatic calculation of the scores. If more than half of the scores in one domain were missing, no result was presented.

For respiratory health, multivariable logistic regression models were used with calculated ORs with 95% CIs. The analyses between first and second/third waves were adjusted for age (≤50, 50–65, >65 years), sex, smoking habits and chronic lung disease. These variables were chosen since they are known as risk factors for a reduced lung function.[29] Associations between patient-related and treatment-related factors and outcomes were adjusted for age (≤65/>65 years), sex, comorbidity, corticosteroids

and invasive ventilation, all known to influence the course of the disease in COVID-19.[30–32] Two separate statistical analyses were conducted: one including participants with at least one test result and a second with participants who had complete data from all three examinations (chest CT scan, pulmonary function test and questionnaires). Statistical analyses were performed with Jamovi V.2.3.19 and an experienced biostatistician validated all analyses.

## Patient and public involvement

Since the hospital was under heavy burden due to COVID-19 patients and members of public health were not involved in the setup of the study.

# RESULTS

## Participants

In total, 678 patients were treated for severe COVID-19 infection during the study period, among which 456 (67%) survived for at least 90 days after ICU discharge. Of these, 278 (61%) were followed up 6 months after ICU discharge (figure 1). There were 182 (72%) who survived from the first wave and 122 (67%) had a follow-up. For the second/third waves, 274 (64%) survived and 156 (57%) had a follow-up. First-wave survivors were younger, had lower BMI, fewer cardiovascular diseases, higher CRP max and were more often treated with invasive ventilation, but less frequently with corticosteroids (table 1). Among survivors with chronic lung disease, the vast majority had asthma. Patient characteristics and ICU-related data were similar between survivors who attended the follow-up and non-participants (online supplemental table 1), but among non-responders who declined follow-up, NIV was

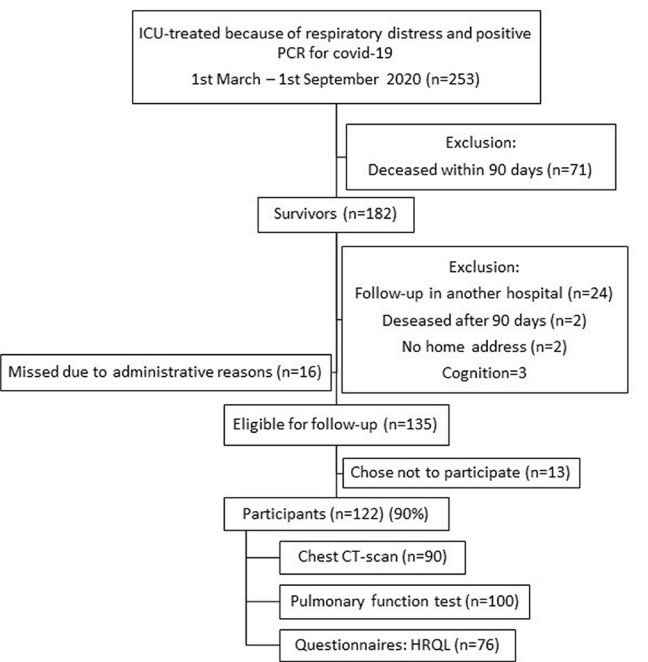
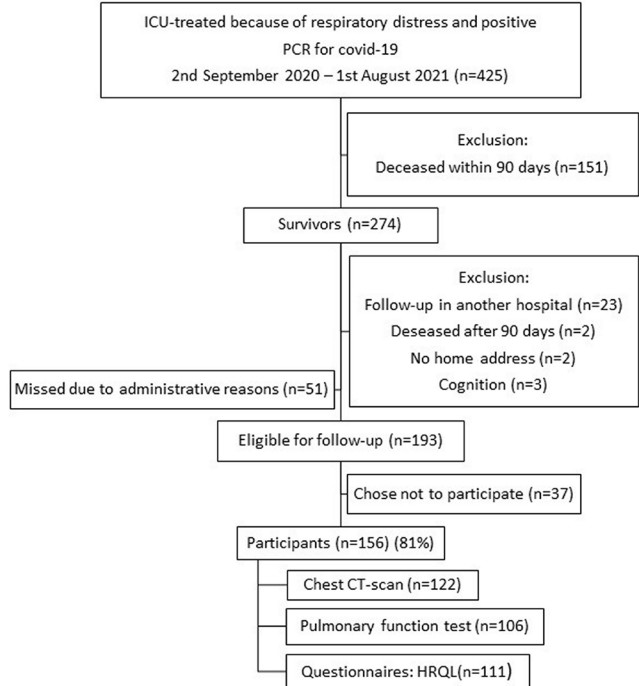

**Figure 1** Flow chart of study inclusion. HRQL, health-related quality of life; ICU, intensive care unit.

**Table 1** Characteristics of included COVID-19 intensive care unit survivors across the first and second/third pandemic waves

| | First wave (1 March 2020–1 September 2020) | Second/ third waves (2 September 2020–1 August 2021) | P value |
|---|---|---|---|
| Total no, n | 122 | 156 | |
| Follow-up time, months, Md (IQR) | 5 (4–5) | 6 (5–8) | <0.001* |
| Age, years Md (IQR) | 58 (50–66) | 61 (53–69) | <0.05* |
| Male n (%) | 93 (76) | 110 (71) | 0.34 |
| BMI, kg/m² Md (IQR) | 28 (25–31) | 29 (26–33) | <0.05* |
| Diabetes n (%) | 24 (20) | 36 (23) | 0.56 |
| Hypertension n (%) | 53 (43) | 72 (46) | 0.72 |
| Cardiovascular disease n (%) | 12 (10) | 30 (19) | <0.05* |
| Chronic lung disease n (%) | 22 (18) | 37 (24) | 0.30 |
| Ever smoker n (%) | 46 (38) | 74 (47) | 0.11 |
| CRP max, mg/L, Md (IQR) | 310 (199–355) | 182 (126–248) | <0.001* |
| SAPS 3, Md (IQR) | 54 (50–60) | 56 (52–59) | 0.52 |
| Invasive ventilation n (%) | 72 (59) | 64 (41) | <0.01* |
| Corticosteroid treatment n (%) | 49 (40) | 156 (100) | <0.001* |
| Length of ICU stay (days), Md (IQR) | 13 (4–23) | 9 (6-16) | 0.33 |

*p<0.05.
BMI, body mass index; CRP, C reactive protein; Md, median; SAPS, Simplified Acute Physiology Score .

more common (online supplemental table 2). Further, distribution of age, sex, comorbidities, corticosteroid treatment and type of ventilation were similar between participants who completed the questionnaires and performed the pulmonary function test and chest CT scan and those who did not (online supplemental table 3).

### Health-related quality of life
Among the 187 ICU survivors who reported HRQL data, 76 were treated in the ICU for COVID-19 during the first wave. First-wave survivors reported lower scores in all HRQL domains, with clinically relevant and statistically significant differences in the role physical domain (MSD 20.2, 95% CI 7.3 to 33.1, p<0.01), bodily pain (MSD 12.2, 95% CI 3.6 to 20.8, p<0.01) and general health (MSD 7.2, 95% CI 0.7 to 13.6, p=0.03) (table 2).

Similar results were found for survivors of COVID-19 with complete case data, except for non-significant results for general health (MSD 6.9, 95% CI –1.3 to 15.2, p=0.1).

In the multivariate analysis, older age (MSD –12.9, 95% CI –21.0 to –4.8, p<0.01) was associated with worse physical function and similar was seen for those with diabetes (MSD –11.0, 95% CI –21.0 to –1.1, p=0.03). Diabetes also

associated with lower scores on general health (MSD –11.6, 95% CI –20.0 to –3.3, p<0.01) and mental health (MSD –9.5, 95% CI –17.3 to –1.7, p=0.02). Survivors treated with invasive ventilation were more likely to report lower scores on bodily pain (MSD –11.8, 95% CI –20.2 to –3.4, p<0.01). Lower scores on role emotional were seen for those with chronic lung disease (MSD –24.2, 95% CI –39.7 to –8.8, p<0.01) (table 3). The univariate analyses can be found in online supplemental table 4. Similar results were seen in the complete-case analysis (ie, participants who had data from all three examinations with questionnaires, chest CT scan and pulmonary function), where also diabetes associated with more bodily pain (MSD –16.4, 95% CI –29.5 to –3.3, p=0.02) but not with lower scores for mental health (MSD –8.9, 95% CI –18.8 to 1.0, p=0.08).

### Pulmonary outcomes
Compared with first wave survivors, patients in the second/third waves had worse pulmonary function with lower DLCO (OR 1.9, 95% CI 1.1 to 3.5, p=0.03) (table 2). Similar findings were seen in patients who had complete data from the three examinations.

Factors associated with worse radiological pulmonary outcome (ie, CT score ≥1) were older age (>65 years) (OR 4.5, 95% CI 1.7 to 11.8, p<0.01), male sex (OR 2.8, 95% CI 1.3 to 6.3, p=0.01) and ICU corticosteroid treatment (OR 4.8, 95% CI 1.3 to 17.0, p=0.02) (table 4).

For the complete-case analysis, only older age was associated with worse CT scores (OR 3.3, 95% CI 1.1 to 10.1, p=0.04).

Worse pulmonary function (DLCO<80%) was associated with older age (OR 1.9, 95% CI 1.0 to 3.6, p=0.04) and female sex (with female as reference) (OR 0.4, 95% CI 0.2 to 0.9, p=0.02) (table 4), but in the complete-case analysis, only female sex was associated with worse capacity for gas exchange (OR 0.3, 95% CI 0.1 to 0.8, p=0.02).

### DISCUSSION
In this prospective cohort study of COVID-19 ICU survivors, HRQL and pulmonary outcomes were relatively similar between the pandemic waves, despite a changing patient casemix, new virus variants and changes in intensive care treatment strategies, except that first wave survivors had better pulmonary function, measured with the capacity for gas exchange (DLCO) and second/third wave survivors reported better role physical, general health and had less bodily pain.

In our study cohort ICU-survivors treated early in the pandemic seemed to have more severe disease since they were younger with a longer stay in the ICU and to a higher extent treated with invasive ventilation. These differences in severity of disease may be due to viral strain[33] and that later in the pandemic corticosteroid treatment was routinely given.[34] The majority were unvaccinated in Sweden during the time for study inclusion.[35] The ICU-survivors later in the pandemic seemed to be more

**Table 2** Health-related quality of life (HRQL) comparing COVID-19 intensive care unit survivors across the first and second/third waves, presented as mean score differences (MSD) with 95% CIs, respectively, ORs with 95% CIs of CT score and diffusion capacity (DLCO)

| HRQL domains | First wave Mean scores (95% CI) | Second/third waves Mean scores (95% CI) | Unadjusted analysis MSD (95% CI) p value | Adjusted analysis MSD (95% CI) p value |
|---|---|---|---|---|
| Total no | 76 | 111 | | |
| Physical function | 62 (57 to 68) | 64 (58 to 69) | 1.3 (−6.7 to 9.3) 0.75 | 3.6 (−4.2 to 11.5) 0.36 |
| Role physical | 31 (22 to 40) | 50 (41 to 58) | 18.5* (5.6 to 31.3) <0.01 | 20.2* (7.3 to 33.1) <0.01 |
| Bodily pain | 59 (52 to 66) | 69 (64 to 75) | 10.8* (2.3 to 19.3) 0.01 | 12.2* (3.6 to 20.8) <0.01 |
| General health | 49 (44 to 54) | 56 (51 to 60) | 6.3 (−0.2 to 12.8) 0.06 | 7.2* (0.7 to 13.6) 0.03 |
| Vitality | 47 (41 to 52) | 53 (48 to 58) | 6.3 (−0.9 to 13.5) 0.09 | 7.0 (−0.2 to 14.2) 0.06 |
| Social function | 62 (56 to 68) | 67 (62 to 73) | 5.6 (−2.9 to 14.0) 0.19 | 6.3 (−2.2 to 14.7) 0.14 |
| Role emotional | 58 (48 to 68) | 68 (60 to 75) | 9.9 (−2.6 to 22.5) 0.12 | 12.0 (−0.5 to 24.6) 0.06 |
| Mental health | 67 (62 to 72) | 74 (70 to 78) | 6.8* (0.8 to 12.8) 0.03 | 6.9 (−0.8 to 13.1) 0.03 |

| | First wave | | Second/third waves | | | |
|---|---|---|---|---|---|---|
| Outcomes | Numbers | Median (IQR) | Numbers | Median (IQR) | OR (95% CI) p value | OR (95% CI) p value |
| CT score | | | | | | |
| 0 | 25 | 0 | 21 | 0 | Reference 1.0 | Reference 1.0 |
| ≥1 | 65 | 7 (5–10) | 101 | 9 (5–13) | 1.8 (1.0 to 3.6) | 1.8 (0.9 to 3.7) |
| | | | | | 0.07 | 0.10 |
| DLCO % of predicted | | | | | | |
| ≥80 | 64 | 96 (90–109) | 49 | 91 (86–96) | Reference 1.0 | Reference 1.0 |
| <80 | 36 | 68 (62–75) | 57 | 65 (53–71) | 2.1 (1.2 to 3.6)* | 1.9 (1.1 to 3.5)* |
| | | | | | 0.01 | 0.03 |

The models were adjusted for age, sex, tobacco smoking, chronic lung disease.
First wave patients' data were used as a reference.

*P value <0.05.
DLCO, diffusing capacity of the lungs for carbon monoxide.

fragile because of older age, higher BMI and cardiovascular disease. Most evidence indicates a small reduction in mortality rates between the first and second/third pandemic waves among critically ill patients.[36 37] However, data about mortality trends during the pandemic vary depending on study time frames, outcomes, casemix and sample size.[2] Only a few studies have investigated whether long-term morbidity follows the mortality results. One

of these studies was a multinational study, in which 1000 patients with COVID-19 and severe hypoxaemia were randomised to either 6 mg or 12 mg dexamethasone, but no differences between groups were found in 3-month mortality or HRQL.[38]

A systematic review and meta-analysis showed that those with more severe initial illness had poorer long-term recovery after COVID-19 infection.[39] In our study,

**Table 3** Multivariate analysis of patient characteristics and clinical characteristics in relation to health-related quality of life in COVID-19 intensive care unit survivors presented as adjusted mean score differences (MSD) with 95% CI

| Variables | Physical function MSD (95% CI) p value | Role physical MSD (95% CI) p value | Bodily pain MSD (95% CI) p value | General health MSD (95% CI) p value | Vitality MSD (95% CI) p value | Social function MSD (95% CI) p value | Role emotional MSD (95% CI) p value | Mental health MSD (95% CI) p value |
|---|---|---|---|---|---|---|---|---|
| **Age, years** | | | | | | | | |
| ≤65 | Reference 1.0 | Reference 1.0 | Reference 1.0 | Reference 1.0 | Reference 1.0 | Reference 1.0 | Reference 1.0 | Reference 1.0 |
| >65 | −12.9* (−21.0 to −4.8) <0.01 | −1.5 (−15.0 to 12.0) 0.83 | −5.3 (−14.0 to 3.5) 0.24 | −2.2 (−9.0 to 4.7) 0.54 | 2.4 (−5.3 to 10.0) 0.54 | −2.6 (−11.6 to 6.3) 0.56 | 1.7 (−11.8 to 14.7) 0.83 | 1.5 (−4.8 to 7.9) 0.64 |
| **Sex** | | | | | | | | |
| Female | Reference 1.0 | Reference 1.0 | Reference 1.0 | Reference 1.0 | Reference 1.0 | Reference 1.0 | Reference 1.0 | Reference 1.0 |
| Male | 5.7 (−3.2 to 14.6) 0.21 | 11.1 (−3.6 to 25.8) 0.14 | 4.7 (−4.8 to 14.2) 0.33 | 4.2 (−3.2 to 11.6) 0.26 | 6.6 (−1.7 to 15.0) 0.12 | 9.8 (−0.1 to 19.5) 0.05 | −4.2 (−18.6 to 10.2) 0.56 | 1.4 (−5.5 to 8.4) 0.69 |
| **Diabetes** | | | | | | | | |
| No | Reference 1.0 | Reference 1.0 | Reference 1.0 | Reference 1.0 | Reference 1.0 | Reference 1.0 | Reference 1.0 | Reference 1.0 |
| Yes | −11.0* (−21.0 to −1.1) 0.03 | −8.5 (−25.1 to 8.1) 0.31 | −9.2 (−19.9 to 1.5) 0.09 | −11.6* (−20.0 to −3.3) <0.01 | −6.3 (−13.2 to 9.9) 0.18 | −5.6 (−16.5 to 5.3) 0.31 | −11.5 (−27.6 to 4.6) 0.16 | −9.5* (−17.3 to −1.7) 0.02 |
| **Hypertension/cardiovascular disease** | | | | | | | | |
| No | Reference 1.0 | Reference 1.0 | Reference 1.0 | Reference 1.0 | Reference 1.0 | Reference 1.0 | Reference 1.0 | Reference 1.0 |
| Yes | −3.2 (−11.5 to 5.3) 0.46 | −1.1 (−15.1 to 13.0) 0.88 | −6.4 (−15.5 to 2.6) 0.16 | 2.4 (−4.7 to 9.4) 0.51 | 1.7 (−6.2 to 9.6) 0.67 | 1.1 (−8.1 to 10.4) 0.81 | 1.2 (−12.6 to 14.9) 0.87 | 3.4 (−3.3 to 1.0) 0.32 |
| **Chronic lung disease** | | | | | | | | |
| No | Reference 1.0 | Reference 1.0 | Reference 1.0 | Reference 1.0 | Reference 1.0 | Reference 1.0 | Reference 1.0 | Reference 1.0 |
| Yes | −5.8 (−15.5 to 3.8) 0.24 | −13.6 (−29.4 to 2.2) 0.09 | −5.6 (−15.9 to 4.6) 0.28 | −6.9 (−14.9 to 1.1) 0.09 | −8.3 (−17.3 to 0.7) 0.07 | −2.9 (−13.4 to 7.4) 0.57 | −24.2* (−39.7 to −8.8) <0.01 | −3.6 (−11.0 to 3.9) 0.35 |
| **Ventilation support** | | | | | | | | |
| HFNO/NIV | Reference 1.0 | Reference 1.0 | Reference 1.0 | Reference 1.0 | Reference 1.0 | Reference 1.0 | Reference 1.0 | Reference 1.0 |
| Invasive ventilation | −6.7 (−14.6 to 1.1) 0.09 | −8.5 (−20.6 to 5.4) 0.2 | −11.8* (−20.2 to −3.4) <0.01 | −1.6 (−8.2 to 4.9) 0.63 | 0.4 (−7.7 to 7.0) 0.93 | −7.5 (−16.1 to 1.0) 0.08 | 0.5 (−12.2 to 13.2) 0.94 | 3.3 (−2.9 to 9.4) 0.30 |
| **In–ICU corticosteroids** | | | | | | | | |
| No | Reference 1.0 | Reference 1.0 | Reference 1.0 | Reference 1.0 | Reference 1.0 | Reference 1.0 | Reference 1.0 | Reference 1.0 |
| Yes | −6.1 (−18.3 to 6.1) 0.33 | −11.9 (−32.4 to 8.5) 0.25 | −4.5 (−17.7 to 8.8) 0.51 | −4.6 (−14.9 to 5.7) 0.38 | −1.7 (−13.2 to 9.9) 0.77 | −0.6 (−14.1 to 12.9) 0.93 | 3.6 (−16.4 to 23.5) 0.72 | 1.3 (−8.3 to 10.9) 0.79 |

All the variables in the univariate analysis were included in the multivariate analysis with adjustment for time during the pandemic.
*Statistically significant p<0.05.
HFNO, high-flow nasal oxygen; ICU, intensive care unit; NIV, non-invasive ventilation.

risk factors for reduced quality of life in at least one parameter at follow-up in ICU survivors treated because of COVID-19 were older age, diabetes, chronic lung disease and invasive ventilation. Diabetes as a risk factor for critical disease and mortality was identified early in the pandemic.[40] Age as a risk factor for severe disease and worse radiological outcome is well known and has been described previously.[9 41] In line with this, severity of disease has in previous literature been reported to be associated with more remaining health related symptoms at follow-up.[9] As previously reported about the first wave survivors in the cohort, their quality of life was reduced compared with the general Swedish population,[8] also consistent with data from other populations.[31]

For pulmonary healing, older age and corticosteroid treatment were associated with more residual changes on chest CT scan. Regarding treatment with corticosteroids, there are still little known about possible harm and long-term effects in COVID-19.[36 42]

Increasing age and female sex were associated with worse lung function at follow-up. In general, there was a predominance of males who become critically ill.[43] The females in need of ICU treatment in our cohort may have had a worse pulmonary impairment before COVID-19 because there was a discrepancy between CT score and the capacity for gas exchange (DLCO) in females. An association between female sex and worse DLCO values in hospitalised patients has also been reported by others.[27]

**Table 4**  Patient characteristics and clinical characteristics in relation to pulmonary outcomes in COVID-19 intensive care unit survivors presented as OR with 95% CIs

| Variables | CT score≥1 | | | DLCO<80% | | |
|---|---|---|---|---|---|---|
| | Numbers | Univariate OR (95% CI) p value | Multivariate OR (95% CI) p value | Numbers | Univariate OR (95% CI) p value | Multivariate OR (95% CI) p value |
| Age, years | | | | | | |
| ≤65 | 141 | Reference 1.0 | Reference 1.0 | 133 | Reference 1.0 | Reference 1.0 |
| >65 | 71 | 4.3 (1.8 to 10.7)* <0.01 | 4.5 (1.7 to 11.8)* <0.01 | 73 | 2.2 (1.2 to 3.9)* <0.01 | 1.9 (1.0 to 3.6)* 0.04 |
| Sex | | | | | | |
| Female | 61 | Reference 1.0 | Reference 1.0 | 57 | Reference 1.0 | Reference 1.0 |
| Male | 151 | 1.8 (0.9 to 3.6) 0.08 | 2.8 (1.3 to 6.3)* 0.01 | 149 | 0.5 (0.3 to 0.9)* 0.02 | 0.4 (0.2 to 0.9)* 0.02 |
| Diabetes | | | | | | |
| No | 166 | Reference 1.0 | Reference 1.0 | 141 | Reference 1.0 | Reference 1.0 |
| Yes | 46 | 1.2 (0.5 to 2.7) 0.69 | 0.8 (0.3 to 2.0) 0.65 | 65 | 1.7 (0.9 to 3.5) 0.12 | 1.5 (0.7 to 3.3) 0.27 |
| Hypertension/cardiovascular disease | | | | | | |
| No | 108 | Reference 1.0 | Reference 1.0 | 103 | Reference 1.0 | Reference 1.0 |
| Yes | 104 | 1.4 (0.7 to 2.7) 0.13 | 1.2 (0.5 to 2.5) 0.72 | 103 | 1.8 (1.0 to 3.2)* 0.04 | 1.5 (0.8 to 2.9) 0.18 |
| Chronic lung disease | | | | | | |
| No | 163 | Reference 1.0 | Reference 1.0 | 162 | Reference 1.0 | Reference 1.0 |
| Yes | 49 | 1.1 (0.5 to 2.4) 0.80 | 1.6 (0.7 to 4.0) 0.30 | 44 | 1.1 (0.6 to 2.2) 0.70 | 0.8 (0.4 to 1.8) 0.69 |
| Ventilation support | | | | | | |
| HFNO/NIV | 105 | Reference 1.0 | Reference 1.0 | 102 | Reference 1.0 | Reference 1.0 |
| Invasive ventilation | 107 | 1.4 (0.7 to 2.8) 0.28 | 1.4 (0.7 to 3.0) 0.32 | 104 | 1.2 (0.7 to 2.0) 0.57 | 1.3 (0.7 to 2.5) 0.35 |
| In-ICU corticosteroids | | | | | | |
| No | 57 | Reference 1.0 | Reference 1.0 | 58 | Reference 1.0 | Reference 1.0 |
| Yes | 155 | 3.0 (1.5 to 6.0)* <0.01 | 4.8 (1.3 to 17.0)* 0.02 | 148 | 2.5 (1.3 to 4.9)* <0.01 | 1.7 (0.7 to 4.2) 0.21 |

Multivariate analysis included all the variables in the univariate analysis and was adjusted for time during the pandemic.

*Statistically significant p<0.05.
DLCO, diffusing capacity of the lungs for carbon monoxide; HFNO, high-flow nasal oxygen; ICU, intensive care unit; NIV, non-invasive ventilation.

### Strength and limitations

The main methodological strengths of the study lie in the prospective and comprehensive data collection together with the use of well-validated outcome tests and questionnaires. The study also has a number of limitations. First, the number of missing for follow-up due to administrative reasons (n=67) may contribute to increase the risk of selection bias. However, it is likely to believe that those patients were missed at random and not systematically. During this time, the hospital was heavily burdened of COVID-19 and all nurses and physicians were needed in direct patient care and eligible patients were sometimes identified after ICU discharge.

Of those who declined follow-up, fewer needed invasive ventilation and they might have reached complete recovery and considered follow-up redundant. Though, when comparing baseline data for all survivors, no statistically significant differences were found between groups. Further, the results may be biased by the higher survival rate in the first wave, which may be explained by changes in patient-mix, viral strain and differences in treatment strategies, and the follow-up rate was higher for those in the first wave. There are also unmeasured possible residual confounders such as experience, bed occupancy and staff. The lack of baseline data prevents identification of new-onset problems. Yet, data on pulmonary function and HRQL are difficult to obtain pre-ICU. Further, the HRQL results were self-reported and may include a risk of response shift. Since HRQL aspects include many variables, and several analyses were conducted, there is also

a risk that some statistically significant results may be due to chance.

The knowledge about long-term outcomes after critical disease of COVID-19 may be applicable for other causes of acute respiratory distress syndrome (ARDS) as well as treatment in the ICU of various conditions where, consistent with our findings, functional impairments are common.[44 45] In addition, to reduce morbidity in ICU patients with COVID-19 infection it is of great importance to investigate what factors may negatively affect recovery.

## CONCLUSION

This study suggests that even though patient casemix has changed, new variants of SARS-CoV-2 have emerged, and intensive care treatment strategies have changed since the start of the pandemic, long-term HRQL and impaired respiratory health remains similar for COVID-19 patients with critical illness. Further, factors associated with severity of disease may play a role for the recovery.

**Author affiliations**
[1]Department of Internal Medicine, Södersjukhuset, Stockholm, Sweden
[2]Department of Clinical Science and Education, Karolinska Institutet, Stockholm, Sweden
[3]Department of Clinical Sciences, Danderyd Hospital, Karolinska Institutet, Section of Anesthesiology and Intensive Care, Danderyds sjukhus, Stockholm, Sweden
[4]Department of Radiology, Södersjukhuset, Stockholm, Sweden
[5]Department of Cardiology, Södersjukhuset, Stockholm, Sweden
[6]Department of Anesthesia and Intensive Care, Södersjukhuset, Stockholm, Sweden

**Acknowledgements** We would like to thank all patients who participated in the study and caregivers involved in the follow-up.

**Contributors** PD and MR contributed to conceptualisation, methodology, formal analysis, investigation, writing—original draft. PD is the guarantor. MC contributed to methodology, investigation, writing—review and editing. GH contributed to methodology, investigation, writing—review and editing. AH: conceptualisation, investigation, writing—review and editing. EJ-A contributed to conceptualisation, formal analysis, writing—review and editing. AS contributed to conceptualisation, methodology, investigation, formal analysis, writing draft—review and editing. All authors read and approved the final manuscript.

**Funding** The authors have not declared a specific grant for this research from any funding agency in the public, commercial or not-for-profit sectors.

**Competing interests** None declared.

**Patient and public involvement** Patients and/or the public were not involved in the design, or conduct, or reporting, or dissemination plans of this research.

**Patient consent for publication** Not applicable.

**Ethics approval** Study approval was granted from the Swedish Ethical Review Authority (2020-03760, 2020-06106 and 2021-02952) and written informed consent was obtained from all included participants.

**Provenance and peer review** Not commissioned; externally peer reviewed.

**Data availability statement** Data are available on reasonable request. The datasets used and/or analysed during the current study are available from the corresponding author on reasonable request.

**ORCID iD**
Pernilla Darlington http://orcid.org/0000-0002-3411-8838

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
