## [Reviewer comments · BMJ Open]

ARTICLE DETAILS

TITLE (PROVISIONAL)	Comparing severe COVID-19 outcomes of first and second/third waves: a prospective single-center cohort study of health-related quality of life and pulmonary outcomes 6 months after infection
AUTHORS	Darlington, Pernilla; Roël, Mari; Cronhjort, Maria; Hanna, Gabriel; Hedman, Anders; Joelsson-Alm, Eva; Schandl, Anna

VERSION 1 – REVIEW

REVIEWER	Griffith, David Royal Infirmary of Edinburgh, Anaesthesia, Critical Care and Pain
REVIEW RETURNED	23-Jan-2023

GENERAL COMMENTS	The authors report the results of an observational study (or service evaluation) comparing outcomes for patients with severe Covid-19 admitted to 2 ICUs in a single hospital in Sweden. They divide a consecutively recruited cohort of patients in 2 according to arbitrarily defined first and second waves and compare functional outcomes between. The 2. In their introduction, the authors note that between pandemic waves, the characteristics and treatment of patients with Covid-19 changed within this institution. Specific mention is made of risk of hospital admission, clinical/scientific understanding of the disease, and “potentially more effective treatments”, but vaccination (another obvious difference) is not mentioned. Although it is slightly difficult to pin down, the research question appears to ask whether functional outcomes have changed between waves, and by implication, they are attempting to test whether changes in some of these differences impacted functional outcomes in severe Covid-19. The major challenge in answering this question is that the patients being admitted to ICU in the second/third waves are likely to be systemically different from patients admitted during the first wave and these differences are likely to have a large impact on long term outcomes. The patients admitted to ICU later in the pandemic are likely to represent vaccine failures, and we know that these patients are very different from the unvaccinated patients admitted during the first wave. Unfortunately, it is very difficult to identify these patients in a dataset or adjust for these systemic differences analytically. Major Comments 1. Because the research question is not clearly defined, it is difficult to establish precisely what the exposure(s) of interest is/are. The article therefore seems to present a single centre service evaluation, testing whether outcomes are improving over
---

	time. The authors would be advised to set out the research question in PECO format. 2. I think there is a second research question pertaining to risk factors for HRQoL (and pulmonary function) in the whole population (ignoring waves) which relates to the data presented in Supplementary Table 3, but the authors do not state this or describe this analysis in the methods. 3. Following on from 1., the paper presents “unadjusted” and “adjusted” comparisons between samples of patients at different time points. In my opinion, these adjustments should be more carefully considered. If the primary research question is “have improvements in treatment of Covid-19 led to changes in physical outcome?”, then it makes sense to ensure that confounders of the relationship between treatment and physical outcome are controlled for. Given that the baseline characteristics and severity of illness of patients changes between waves due to vaccination, strain, then it makes sense to adjust for baseline characteristics (only age, sex, chronic lung disease and tobacco smoking are adjusted for), but I don’t think it makes sense to adjust for treatments (e.g. corticosteroid treatment or invasive ventilation (p40 line 10)). 4. The dichotomisation of the cohort is not fully justified. It hinges on what the definition of a Covid-19 wave is. This requires more clarification and referencing. Is there a standard definition of a pandemic wave, and does this study adhere to it? 5. The authors need to explain the impact of vaccination on their analysis as it is not discussed. In the second cohort, the widespread uptake of vaccination would lower mortality from Covid-19, so more (potentially frailer) will survive to 90 days. 6. The methods section does not clearly identify the outcome measures of interest. Although SF-36v2 is identified as the measure of HRQoL, the authors report individual domains, but not the component score (PCS and MCS) or the overall score (SF-36). Minor Comments 1. Please give some information about admission criteria to ICU in your hospital. In some hospitals in the UK, NIV and HFO2 were provided in a non-ICU setting and patients were escalated to ICU where this failed. Could the authors describe their local setup. 2. Confounders – please include a short section explaining how confounders were considered in the study. 3. SF-36v2 – was this calibrated to Swedish or US population normative data. 4. A small point perhaps, but patient and public involvement is important, but not always possible. Can the team explain why patients and members of public were not involved in this study. 5. Why have you used age categorisation rather than continuous data. Use of continuous data is generally preferred to prevent unnecessary loss of statistical power. 6. Were survival and follow up rates between waves similar? 7. Follow up rate is low. Of the 456 patients that survived, only 187 patients (41%) reported SF-36 and 212 (52%) had CT scans, and 206 (46%) had DLCO% measured. Although the supplementary tables suggest that participants were broadly similar to non-participants, more information about these 2 groups would be helpful particularly relating to variables known to affect outcome (e.g. pre-morbid comorbidity is strongly associated with SF-36v2).
REVIEWER	Rosa, Regis Goulart

	HMV, Intensive Care
REVIEW RETURNED	25-Jan-2023

GENERAL COMMENTS	The present study aimed to clarify whether outcomes in ICU survivors treated for severe COVID-19 infection have improved with time and to identify factors associated with worse long-term outcomes. The research question is interesting and the study seems to have been well conducted. Please find below some suggestions/commentaries. 1) Study design and setting - I might suggest to describe the main SARS-CoV-2 variants responsible for the studied COVID-19 waves in Stockholm. Information on COVID-19 vaccination coverage in Stockholm during the second/third wave would be informative as well. Please also discuss the potential impact of these factors on study results. 2) Methods - Study design and setting: Please describe the characteristics of the ICU in order to allow readers to better evaluate external validity (e.g., Number of beds, proportion of patients per ICU staff [physician, nurse, physiotherapist, psychologist etc], age range of admitted patients). Was the ICU exclusive for COVID-19 patients? Were ICU resources comparable across different waves? 3) Methods - Participants: Please describe the criteria used to exclude participants from main analyses in order to be consistent with figure 1. 4) Statistical analyses - Please describe how the sample size was calculated. 5) Statistical analyses - Please describe the rationale for choosing age, sex, comorbidity, tobacco smoking habits, corticosteroids, and invasive mechanical ventilation as covariates in multivariable models. 6) The 90 day mortality during the second/third waves was higher than that of first wave. Please comment the possible impact of survival bias as a potential explanation for the study results. 7) Was quality of life of ICU COVID-19 survivors comparable to the age- and sex-matched general population in Sweden? 8) Please comment how the present study results on quality of life relate to other long-term follow-up studies with COVID-19 survivors (e.g., DOI: 10.1007/s00134-022-06953-1) and non-COVID-19 ICU survivors (e.g., DOI: 10.1056/NEJMoa1011802 and DOI:10.1007/s00134-015-3669-5). 8) Please include the high number of exclusions as an important limitation.
---

VERSION 1 – AUTHOR RESPONSE

Reviewer: 1
Dr. David Griffith, Royal Infirmary of Edinburgh

Comments to the Author:

The authors report the results of an observational study (or service evaluation) comparing outcomes for patients with severe Covid-19 admitted to 2 ICUs in a single hospital in Sweden. They divide a consecutively recruited cohort of patients in 2 according to arbitrarily defined first and second waves and compare functional outcomes between. The 2. In their introduction, the authors note that between pandemic waves, the characteristics and treatment of patients with Covid-19 changed within this institution. Specific mention is made of risk of hospital admission, clinical/scientific understanding of the disease, and “potentially more effective treatments”, but vaccination (another obvious difference) is not mentioned. Although it is slightly difficult to pin down, the research question appears to ask whether functional outcomes have changed between waves, and by implication, they are attempting to test whether changes in some of these differences impacted functional outcomes in severe Covid-19.

The major challenge in answering this question is that the patients being admitted to ICU in the second/third waves are likely to be systemically different from patients admitted during the first wave and these differences are likely to have a large impact on long term outcomes. The patients admitted to ICU later in the pandemic are likely to represent vaccine failures, and we know that these patients are very different from the unvaccinated patients admitted during the first wave. Unfortunately, it is very difficult to identify these patients in a dataset or adjust for these systemic differences analytically.

Major Comments

1. Because the research question is not clearly defined, it is difficult to establish precisely what the exposure(s) of interest is/are. The article therefore seems to present a single centre service evaluation, testing whether outcomes are improving over time. The authors would be advised to set out the research question in PECO format.

Reply: We appreciate this suggestion and have now written in the abstract "To clarify if outcomes have improved with time in intensive care unit (ICU) survivors treated for severe COVID-19, comparing first with second/third waves ." and at the end of the Introduction "Therefore, this prospective single-center cohort study aimed to clarify whether in ICU survivors treated for severe COVID-19 infection comparing first with second/third waves if outcomes have improved with time. Further, to identify factors associated with worse long-term outcomes."

2. I think there is a second research question pertaining to risk factors for HRQoL (and pulmonary function) in the whole population (ignoring waves) which relates to the data presented in Supplementary Table 3, but the authors do not state this or describe this analysis in the methods.

Reply: We apologize and have now added to Methods "Further to investigate potential factor's role for the long-term outcome for HRQL and pulmonary impairment following were included: age, sex, comorbidities (i.e diabetes, hypertension/cardiovascular disease, chronic lung disease), ventilation support and in ICU-corticosteroids. All known to influence the course of the disease in COVID-19". Ref <https://doi.org/10.1186/s12879-021-06536-3>, doi: 10.1056/NEJMoa2021436, DOI: 10.1007/s00134-022-06953-1. We have also added diabetes to the analysis.

3. Following on from 1., the paper presents “unadjusted” and “adjusted” comparisons between samples of patients at different time points. In my opinion, these adjustments should be more carefully considered. If the primary research question is “have improvements in treatment of Covid-19 led to changes in physical outcome?”, then it makes sense to ensure that confounders of the relationship between treatment and physical outcome are controlled for. Given that the baseline characteristics and severity of illness of patients changes between waves due to vaccination, strain, then it makes

sense to adjust for baseline characteristics (only age, sex, chronic lung disease and tobacco smoking are adjusted for), but I don't think it makes sense to adjust for treatments (e.g. corticosteroid treatment or invasive ventilation (p40 line 10)).

Reply: We agree with the reviewer and have recalculated the adjusted models as suggested.

4. The dichotomisation of the cohort is not fully justified. It hinges on what the definition of a Covid-19 wave is. This requires more clarification and referencing. Is there a standard definition of a pandemic wave, and does this study adhere to it?

Reply: This is a valid comment. We have now added ref Zhang et al, 2021, A second wave? What do people mean by COVID waves? – A working definition of epidemic waves. In the paper they define an epidemic wave as constituting of some upward and/or downward periods where the increase in an upward period has to be substantial by sustaining over a period of time to distinguish them from an uptick, a downtick, reporting errors, or volatility in new cases. We have also added ref Ludvigsson, 2022, How Sweden approached the COVID-19 pandemic: Summary and commentary on the National Commission Inquiry. We have rewritten "In Sweden, the first cases of severely ill patients were admitted to the ICU at the beginning of March 2020 with a peak in April and then subsided during the summer (ref. Ludvigsson). Therefore, this time period has been considered to separate the end of the first wave (ref Zhang et al). The beginning of the second wave was estimated to start on 2 September 2020. Thereafter under a longer period, there were an increase in cases with one peak around new year and another peak in April 2021. There was no clear gap in time between the second and third waves, even if there was a clear decrease in cases in March 2021, and the second and third waves were therefore grouped as one time period in this study."

5. The authors need to explain the impact of vaccination on their analysis as it is not discussed. In the second cohort, the widespread uptake of vaccination would lower mortality from Covid-19, so more (potentially frailer) will survive to 90 days.

Reply: We understand the reviewer's concern. In Stockholm, Sweden, there were delays in the vaccination program and when it started, those who were too fragile to be considered for intensive care treatment such as those in nursing homes were the first to be vaccinated. For those who would be considered for intensive care treatment the vaccination started in March/April 2021, with again the oldest first and this was around the peak of the third wave,, therefore we believe the vaccination against COVID-19 had little impact on the results. We have now added to the Methods "Since the vaccination against COVID-19 in Stockholm started with the oldest and most fragile first and for adults ≥ 75 at the end of March close to the peak of the third wave (ref Isitt et al, The early impact of vaccination against SARS-CoV-2 in Region Stockholm Sweden), this is believed to have little impact on the cohort."

6. The methods section does not clearly identify the outcome measures of interest. Although SF-36v2 is identified as the measure of HRQoL, the authors report individual domains, but not the component score (PCS and MCS) or the overall score (SF-36).

Reply: In this study, we used RAND-36 to assess HRQL. RAND-36 and SF-36 are similar, except for some minor differences, such as that RAND-36 lacks an authorized algorithm for calculating Mental and Physical Component Summary scores. Therefore, we focused on presenting the results for each HRQL sub-domain. We have added this to the paragraph about outcome measures.

Reference: Orwelius L, et al. The Swedish RAND-36 Health Survey – reliability and responsiveness assessed in patient populations using Svensson's method for paired ordinal data. J Patient Rep Outcomes, 2018 Dec 2:4.

Minor Comments

1. Please give some information about admission criteria to ICU in your hospital. In some hospitals in the UK, NIV and HFNO were provided in a non-ICU setting and patients were escalated to ICU where this failed. Could the authors describe their local setup.

Reply: This is an adequate question. The admission criteria were need of respiratory support. During the first wave all patients with NIV and HFNO were treated in the ICU, during the second/third wave HFNO treatment were initiated in the ICU and when judged to be stabilized continued outside the ICU but under continues observation at a specialized ward with this purpose as a step down. We have now to the method section added “The admission criteria to the ICU were need of respiratory support (ie HFNO/NIV/invasive ventilation). During second/third wave there was a step down for stable patients on HFNO, but still under continues observation. This whole period was included in the length of ICU stay.”

2. Confounders – please include a short section explaining how confounders were considered in the study.

Reply: Thank you for this comment. We have added to Methods “Further to investigate potential factor’s role for the long-term outcome for HRQL and pulmonary impairment, following were included: age, sex, comorbidities (i.e. diabetes, hypertension/cardiovascular disease, chronic lung disease), ventilation support and in ICU-corticosteroids. All of them known to influence the course of the disease in COVID-19. Ref <https://doi.org/10.1186/s12879-021-06536-3>, doi: 10.1056/NEJMoa2021436, DOI: 10.1007/s00134-022-06953-1. We have added diabetes as well as a confounder. Other conditions such as immunosuppressive disorders and cancer were only rarely seen an therefore not included. Obesity was not included since we in a previous study saw that they tended to have relatively little residual pulmonary changes doi.org/10.1111/crj.13453 and likely to have had respiratory failure due to factors associated with their overweight. Further we have added to Statistical analyses “For the multivariate analysis comparing waves, confounders other than age (≤ 50 , 50-65, >65 years) and sex (male/female), in this case tobacco smoking habits (ever/never) and chronic lung disease (yes/no) were chosen since they are known as risk factors for a reduced lung function.” doi: 10.1183/09059180.00003609.

3. SF-36v2 – was this calibrated to Swedish or US population normative data.

Reply: Psychometric properties of the Swedish version of RAND-36 have been evaluated in a Swedish general population. The performance of the instrument was tested in the total sample as well as in subgroups by gender, age, education and occupation. The completeness of data was satisfactory at item level as well as at the scale level, indicating that the questionnaire was well accepted. The study also provided reference data for norm-based comparisons. Ohlsson-Nevo E., et al. The Swedish RAND-36 : psychometric characteristics and reference data from the Mid-Swed Health Survey. *J Patient Rep Outcomes*, 2021, Dec 5:66. doi: 10.1186/s41687-021-00331-z We have added this to the paragraph about outcome measures.

4. A small point perhaps, but patient and public involvement is important, but not always possible. Can the team explain why patients and members of public were not involved in this study.

Reply: This is a good point. Have now added to Patient and public involvement “Since the hospital was under heavy burden due to COVID-19 and the follow-up had to be organised within a short time,

not knowing what to expect, patients and members of public health were not involved in the set-up of the study.”

5. Why have you used age categorisation rather than continuous data. Use of continuous data is generally preferred to prevent unnecessary loss of statistical power.

Reply: We have carefully considered the reviewer’s comments about dichotomization of covariates. We discussed this with our biostatistician and agreed on the following categories for age <50, 50-65, >65 years (a common internationally established categorization).

6. Were survival and follow up rates between waves similar?

Reply: This is an adequate question about selection bias. We have added to limitations that “Further the results may be biased by the higher survival rate in the first wave which could have several possible explanations such as change in patient-mix, viral strain, hospital load and differences in treatment strategies, and the follow-up rate was also higher for those in the first wave.” Further we have added to Strengths and limitations page 3 “Further survival rate and follow-up rate were higher for those from the first wave.”

7. Follow up rate is low. Of the 456 patients that survived, only 187 patients (41%) reported SF-36 and 212 (52%) had CT scans, and 206 (46%) had DLCO% measured. Although the supplementary tables suggest that participants were broadly similar to non-participants, more information about these 2 groups would be helpful particularly relating to variables known to affect outcome (e.g. pre-morbid comorbidity is strongly associated with SF-36v2).

Reply: We understand the reviewers concern and analysis of missing data has been performed among those who attended the follow-up (see supplementary table) to investigate the risk of non-response bias among those who did not complete the questionnaires or perform the CT-scan or the spirometry test. No statistically significant differences between the groups were found. We have added to the Results “Further, distribution of age, sex, comorbidities, corticosteroid treatment and type of ventilation were similar between participants who completed the questionnaires and performed the pulmonary function test and chest CT scan and those who did not (Supplementary Table 3).” We have also added “Median time to follow-up with chest CT scan was 6 months (IQR 5-7) and for pulmonary function test 6 months (IQR 6-7), with no significant differences between waves.” Further, we have added to limitations “Firstly, the high number of exclusions due to administrative reasons and as well referral to other hospitals due to lack of beds in the ICU and with own follow-up programs are also to be considered. During this time, the hospital was under heavy burden and all ICU nurses and physicians were needed in bedside work and the follow-up had to be done retrospectively. However, those missed due to administrative reasons were most likely missed at random.”

Reviewer: 2

Dr. Regis Goulart Rosa, HMG

Comments to the Author:

The present study aimed to clarify whether outcomes in ICU survivors treated for severe COVID-19 infection have improved with time and to identify factors associated with worse long-term outcomes. The research question is interesting and the study seems to have been well conducted. Please find below some suggestions/commentaries.

1) Study design and setting - I might suggest to describe the main SARS-CoV-2 variants responsible for the studied COVID-19 waves in Stockholm. Information on COVID-19 vaccination coverage in Stockholm during the second/third wave would be informative as well. Please also discuss the potential impact of these factors on study results.

Reply: This is an adequate question also addressed by the other reviewer. In the beginning of the pandemic in Sweden different virus strains were not tested for, during the second wave a non-variant of concern (VOC) were dominating and during the third wave the Alpha-VOC was dominating, the Alpha-VOC cohort had in a Swedish study higher rates of severe illness among hospitalized patients (ref Impact of the Alpha VOC on disease severity in SARS-CoV-2 positive adults in Sweden). Regarding vaccination, in Stockholm, Sweden, there were delays in the vaccination programme and when it started, those who were too fragile to be considered for intensive care treatment such as those in nursing homes were the first to be vaccinated. For those who would be considered for intensive care treatment the vaccination started in March/April 2021, with again the oldest first and this was around the peak of the third wave,, therefore we believe the vaccination against COVID-19 had little impact on the results. We have now added to the Methods “Since the vaccination against COVID-19 in Stockholm started with the oldest and most fragile first and for adults ≥ 75 at the end of March close to the peak of the third wave (ref Isitt et al, The early impact of vaccination against SARS-CoV-2 in Region Stockholm Sweden), this is believed to have little impact on the cohort.”

2) Methods - Study design and setting: Please describe the characteristics of the ICU in order to allow readers to better evaluate external validity (e.g., Number of beds, proportion of patients per ICU staff [physician, nurse, physiotherapist, psychologist etc], age range of admitted patients). Was the ICU exclusive for COVID-19 patients? Were ICU resources comparable across different waves?

Reply: We have now added to Methods “Before the pandemic, the two ICUs had a total number of 16 beds. During the first wave, the number of ICU beds expanded to a maximum of 60 beds in April and May 2020. In the beginning of the second wave, the number of ICU beds were 21, rising to 29 beds in November 2020 and further up to maximum 33 beds in April and May 2021. At the end of the third wave, the number of beds were 21. The nurse:patient ratio was 1:1-2 and physician:patient ratio 1:4-6 during all three waves, but the proportion of intensive care specialised nurses and physicians decreased during the peaks of the pandemic. None of the ICUs were exclusive for Covid-19 patients.”

3) Methods - Participants: Please describe the criteria used to exclude participants from main analyses in order to be consistent with figure 1.

Reply: We apologize and have now added to Methods “The exclusion criteria as showed in Figure 1 were diseased within 90 days after ICU admission, follow-up in another hospital being referred to or judged not to be able to participate (i.e. no home address or impaired cognition).” Have now lift out “Missed due to administrative reasons” and put it separate.

4) Statistical analyses - Please describe how the sample size was calculated.

Reply: We have added to Statistical analysis “Since the study was part of a follow-up program after ICU-treatment with extra investigations for those with ARDS due to COVID-19 during the first wave and later extended to also include second/third wave survivors, the sample size was not calculated in advance”.

5) Statistical analyses - Please describe the rationale for choosing age, sex, comorbidity, tobacco smoking habits, corticosteroids, and invasive mechanical ventilation as covariates in multivariable models.

Reply: Thank you for this comment addressed also by the other reviewer. As suggested by the other reviewer we have from the multivariate analysis comparing waves removed corticosteroid treatment and invasive ventilation. We have added to Methods “Further to investigate potential factor’s role for the long-term outcome for HRQL and pulmonary impairment, following were included: age, sex, comorbidities (i.e. diabetes, hypertension/cardiovascular disease, chronic lung disease), ventilation support and in ICU-corticosteroids. All of them known to influence the course of the disease in COVID-19. Ref <https://doi.org/10.1186/s12879-021-06536-3>, doi: 10.1056/NEJMoa2021436, DOI: 10.1007/s00134-022-06953-1. We have also added diabetes to the analysis. Other conditions such as immunosuppressive disorders and cancer were only rarely seen and therefore not included. Obesity was not included since we in a previous study saw that they tended to have relatively little residual pulmonary changes doi.org/10.1111/crj.13453 and likely had respiratory failure due to factors associated with their overweight.

Further we have added to Statistical analyses “For the multivariate analysis comparing waves, confounders other than age (≤ 50 , 50-65, >65 years) and sex (male/female), in this case tobacco smoking habits (ever/never) and chronic lung disease (yes/no) were chosen since they are known as risk factors for a reduced lung function.” doi: 10.1183/09059180.00003609.

6) The 90 day mortality during the second/third waves was higher than that of first wave. Please comment the possible impact of survival bias as a potential explanation for the study results.

Reply: This is an adequate question about selection bias also addressed by the other reviewer. We have added to limitations that “Further the results may be biased by the higher survival rate in the first wave which could possibly be explained by change in patient-mix, viral strain, hospital load and differences in treatment strategies, and also the follow-up rate was higher for those in the first wave.” Further we have added to Strengths and limitations page 3 “Further survival rate and follow-up rate were higher for those from the first wave.”

7) Was quality of life of ICU COVID-19 survivors comparable to the age- and sex-matched general population in Sweden?

Reply: This is a good point. We have in a previous follow-up study at about five months of ICU-survivors with COVID-19 from the first wave performed a comparison with an age- and sex-matched general population in Sweden and found that ICU survivors treated because of COVID-19 had lower scores in all domains doi: 10.1111/aas.13939. We have now added this information to the Discussion “Our findings as previously reported with a reduced health quality-of-life in the most severely ill COVID-19 survivors compared with the general population (ref above Schandl et al) are also consistent with data from other populations (suggested ref below DOI: 10.1007/s00134-022-06953-1).

8) Please comment how the present study results on quality of life relate to other long-term follow-up studies with COVID-19 survivors (e.g., DOI: 10.1007/s00134-022-06953-1) and non-COVID-19 ICU survivors (e.g. DOI: 10.1056/NEJMoa1011802 and DOI:10.1007/s00134-015-3669-5).

Reply: We thank the reviewer for this suggestion. We have added the first reference as answered above. We have also added the second and third reference to the Discussion and written “This knowledge may also be applicable for other causes of acute respiratory distress syndrome (ARDS) as well as treatment in the ICU of various conditions where, consistent with our findings, functional impairments are common (second and third suggested ref).”

8) Please include the high number of exclusions as an important limitation

Reply: We agree with the reviewer about this concern and have added to limitations “Firstly, the high number of exclusions due to administrative reasons and as well referral to other hospitals due to lack

of beds in the ICU and with own follow-up programs are also to be considered. During this time, the hospital was under heavy burden and all ICU nurses and physicians were needed in bedside work and the follow-up had to be done retrospectively. However, those missed due to administrative reasons were most likely missed at random.”

VERSION 2 – REVIEW

REVIEWER	Griffith, David Royal Infirmary of Edinburgh, Anaesthesia, Critical Care and Pain
REVIEW RETURNED	10-Mar-2023

GENERAL COMMENTS	Overall, whilst efforts have been made in the direction of the previous comments, I think the manuscript still lacks clarity in terms of aims, research questions, and methods. I would advise the authors to revisit my previous comments and consider whether they can address these further. The interest in noting that outcomes change between first wave and second/third waves is limited unless there is also an attempt to understand why this is – is it because the patients are different? (e.g. due to vaccination, escalation policy). Is it because the disease is different? (e.g. due to a different viral strain). Is it because we are giving everyone steroids and delaying IPPV in favour of NIV? Or are there unmeasured factors (experience, bed occupancy, staff) that might persevere after adjusting for everything else? This analysis misses the opportunity to answer or discuss these questions. I don't think the manuscript has been changed much from the previous version, but the response to reviewer comments have perhaps clarified the goals of the work. I would recommend improving the discussion around some of the unanswered questions from the work. I would also recommend extreme caution around the main findings which hinge on a p value threshold of 0.05 which given the huge number of comparisons (19 multivariable models each including 7 covariates) must surely be questioned.
--

REVIEWER	Rosa, Regis Goulart HMV, Intensive Care
REVIEW RETURNED	31-Mar-2023

GENERAL COMMENTS	I have no additional suggestions.
-----------------------------------

VERSION 2 – AUTHOR RESPONSE

Reviewer: 1

Dr. David Griffith, Royal Infirmary of Edinburgh

Comment:

Overall, whilst efforts have been made in the direction of the previous comments, I think the manuscript still lacks clarity in terms of aims, research questions, and methods. I would advise the authors to revisit my previous comments and consider whether they can address these further.

Reply: We have now rewritten the research question as suggested according to the PECO-format. The Method section has been reorganized and rewritten including the following subheadings: study design and participants, setting, data collection, pandemic waves (i.e. exposure, including references for definition of waves), outcomes (health-related questionnaires, chest CT scan and pulmonary function) and statistical analyses (including information about associations between patient- and treatment-related factors and what outcomes here were adjusted for with references).

Comment: The interest in noting that outcomes change between first wave and second/third waves is limited unless there is also an attempt to understand why this is – is it because the patients are different? (e.g. due to vaccination, escalation policy). Is it because the disease is different? (e.g. due to a different viral strain). Is it because we are giving everyone steroids and delaying IPPV in favour of NIV? Or are there unmeasured factors (experience, bed occupancy, staff) that might persevere after adjusting for everything else? This analysis misses the opportunity to answer or discuss these questions.

Reply: We thank the reviewer for the efforts to further improve our manuscript. As suggested, we have revised the discussion and addressed what has been pointed out and in favor of this shortened other parts. We have also added a discussion about potential role of viral strain, vaccination, and corticosteroid treatment. In the limitations we have further addressed the risk of residual confounding.

Comment: I don't think the manuscript has been changed much from the previous version, but the response to reviewer comments have perhaps clarified the goals of the work. I would recommend improving the discussion around some of the unanswered questions from the work. I would also recommend extreme caution around the main findings which hinge on a p value threshold of 0.05 which given the huge number of comparisons (19 multivariable models each including 7 covariates) must surely be questioned.

Reply: We agree with the reviewer and have included as a limitation the following sentence about the risk with mass-significant testing: "Since HRQL aspects include many variables, and several analyses were conducted, there is also a risk that some statistically significant results may be due to chance." Further, factors known to be associated with severity of disease (Gold et al COVID-19 and comorbidities: a systematic review and meta-analysis) seemed in our study to be associated with a reduced health-quality of life, and also in line with this the severity of disease has previously shown to be associated with more fatigue or muscle weakness at follow-up as well as anxiety or depression (Huang et al 6-month consequences of COVID-19 in patients discharged from hospital: a cohort study). We have therefore added the following sentence to the discussion where HRQL aspects are mentioned: "In line with this, severity of disease has in previous literature been reported to be associated with more remaining health-related symptoms at follow-up". We have further, after advice from a biostatistician, added p-values to facilitate the interpretation regarding statistical power.

With these changes we do believe that the manuscript has improved and hope that the revised version can be accepted for publication in BMJ Open.

Sincerely,

For the authors,
Pernilla Darlington
Respiratory Medicine Division, Department of Medicine,
Södersjukhuset
E-mail: pernilla.darlington-lidin@regionstockholm.se